# An Adaptive RBF-NMPC Architecture for Trajectory Tracking Control of Underwater Vehicles

**Zhenzhong Chu** [1,*], **Da Wang** [2] **and Fei Meng** [3]

1    Logistics Engineering College, Shanghai Maritime University, Shanghai 201306, China
2    College of Information Engineering, Shanghai Maritime University, Shanghai 201306, China; wangda0030@stu.shmtu.edu.cn
3    Department of System Science, University of Shanghai for Science and Technology, Shanghai 201306, China; feimeng@usst.edu.cn
*    Correspondence: zzchu@shmtu.edu.cn; Tel.: +86-021-3828-2643

**Abstract:** An adaptive control algorithm based on the RBF neural network (RBFNN) and nonlinear model predictive control (NMPC) is discussed for underwater vehicle trajectory tracking control. Firstly, in the off-line phase, the improved adaptive Levenberg–Marquardt-error surface compensation (IALM-ESC) algorithm is used to establish the RBFNN prediction model. In the real-time control phase, using the characteristic that the system output will change with the external environment interference, the network parameters are adjusted by using the error between the system output and the network prediction output to adapt to the complex and uncertain working environment. This provides an accurate and real-time prediction model for model predictive control (MPC). For optimization, an improved adaptive gray wolf optimization (AGWO) algorithm is proposed to obtain the trajectory tracking control law. Finally, the tracking control performance of the proposed algorithm is verified by simulation. The simulation results show that the proposed RBF-NMPC can not only achieve the same level of real-time performance as the linear model predictive control (LMPC) but also has a superior anti-interference ability. Compared with LMPC, the tracking performance of RBF-NMPC is improved by at least 43% and 25% in the case of no interference and interference, respectively.

**Keywords:** underwater vehicle; trajectory tracking; neural networks; nonlinear model predictive control

## 1. Introduction

With the progress of intelligent control technology, the development of underwater vehicles has entered a new stage. Whether it is the exploration and exploitation of marine mineral resources, the investigation of marine topography, or military applications, it is inseparable from the participation of underwater vehicles [1]. Trajectory tracking is one of the key technologies in the field of underwater vehicles. It is the premise and guarantee for underwater vehicles to complete the specified tasks [2]. Therefore, the research of trajectory tracking control technology is particularly important.

At present, the main research methods of underwater vehicle trajectory tracking control are proportional-integral-derivative (PID) control, fuzzy control, backstepping control, sliding mode control, etc. In [3], a variable integral PID controller based on disturbance observer has been designed to realize the heading control of an underwater vehicle. In [4], good trajectory tracking results have been achieved for underactuated underwater vehicles using terminal sliding mode control. In [5], a bio-inspired backstepping control method and a three-dimensional trajectory tracking controller have been proposed for the deep-diving control. However, most of these methods do not consider the constraints of system state and input, which leads to the phenomenon of thrust saturation in actual control easily. On the contrary, model predictive control (MPC) has the ability to deal with various constraints and has great flexibility in describing control problems. These remarkable

features caused the MPC algorithm to gradually become a hot research topic in trajectory tracking control [6].

However, the coupled, nonlinear, time-varying dynamics characteristic of underwater vehicles make the design of MPC very challenging. The nonlinear effects of hydrodynamic damping, Coriolis, and centripetal forces induce two main difficulties in using the nonlinear model predictive control (NMPC) when the underwater vehicle is working.

Firstly, it is difficult to establish an accurate model of the underwater vehicle. The environmental disturbance represented by the ocean current, load variation and change of hydrodynamic parameters will affect the establishment of the dynamic model of the underwater vehicle; in addition, the commonly used dynamic model of the underwater vehicle established by the Newton–Euler equation is a parametric model, which does not have the function of online adaptive correction [7]. It is difficult to ensure accuracy and applicability when the underwater vehicle is in a complex environment. The neural network has a wide application in underwater vehicle modeling [8–10] because of its good nonlinear approximation ability and adaptive learning function. The RBFNN is used in [11] to approximate the uncertain interference and the uncertainty of the AUV model to suppress the influence of parameter perturbation. In [12], an adaptive sliding mode control strategy based on RBFNN was proposed to solve the heading control problem of AUV.

Secondly, it is difficult to guarantee the real-time performance of the optimization process. The real-time performance of MPC is affected by dynamic model complexity and the rolling optimization algorithm [13]. At present, linear model predictive control (LMPC) is often used to simplify the nonlinear model in order to meet the real-time requirements. In [14], combined the LMPC with sonar images, the dynamic target tracking problem is realized. In [15], the complex six degrees of freedom underwater vehicle mathematical model is linearized, and smooth tracking between trajectory points is achieved. In [16], the linear robust MPC is applied to ensure the stability of surface ships in disturbance environments. Linearization technology plays an important role in the analysis and design of NMPC. However, the frequent disturbance of the ocean current and the complexity and nonlinearity of the dynamic model degrade the control performance of linearization considerably. When the curvature of the reference trajectory changes greatly, the LMPC is prone to overshoot [17]. In addition, it is difficult to guarantee the feasibility of optimizing the problem at each sampling time because the linearization method changes the model online. Therefore, how to establish an accurate nonlinear model while ensuring real-time performance is particularly important in the development of NMPC.

In this paper, a trajectory tracking control architecture combined with NMPC and RBFNN is studied. In the off-line phase, the random step signals are used as the excitation signal to obtain the state response of the underwater vehicle, which are taken as the model training sample. The improved adaptive Levenberg–Marquardt-error surface compensation (IALM-ESC) algorithm is applied to the underwater vehicle system identification, and fewer network nodes are used to reflect the dynamic characteristics. In the real-time control phase, the system output collected in each sampling period contains the information of external interference. The network parameters are updated online according to the error between the system output and the network prediction output. At the same time, an improved adaptive gray wolf optimization (AGWO) is proposed to improve the NMPC optimization performance and ensure the real-time control. The innovation of this paper mainly includes the following aspects:

(1) The structure and parameters of the radial basis function neural network (RBFNN) are determined by the IALM-ESC algorithm. Compared with the traditional gradient descent (GD) method, the applied algorithm has great improvement in convergence speed and convergence effect.

(2) In the real-time control stage, the neural network parameters are adjusted and updated online according to the prediction error, which improves the adaptive ability of the controller in the complex underwater environment.

(3) Based on the traditional gray wolf optimization (GWO) algorithm, the idea of adaptive weight and worst-case crossover is added to improve the global search ability and convergence speed, so as to ensure the real-time performance of NMPC.

The content of this paper is arranged as follows: Section 2 introduces the problems of this paper, including the kinematics and dynamics model of underwater vehicles. Section 3 introduces the design process of the controller. Section 4 is the related simulation experiments, including the results of model identification, optimization and tracking control. Some results and future work are described in Section 5.

## 2. Problem Description

This paper studies the trajectory tracking control of the underwater vehicles in the horizontal plane, and the motion of surge, sway and yaw are considered.

The kinematics model of underwater vehicles in the horizontal plane is shown in (1), which is used to describe the transformation relationship between the motion coordinate system and inertial coordinate system.

$$\dot{\eta} = J(\eta)v \tag{1}$$

where $J(\eta)$ is the coordinate transformation matrix, which is defined as:

$$J(\eta) = \begin{bmatrix} \cos\psi & -\sin\psi & 0 \\ \sin\psi & \cos\psi & 0 \\ 0 & 0 & 1 \end{bmatrix} \tag{2}$$

where $\psi$ is the heading angle.

The dynamic model of underwater vehicles is represented as [18]:

$$M\dot{v} + C(v)v + D(v)v + g(\eta) = \tau \tag{3}$$

where $v = [u, v, r]^T$ is the vector composed of surge velocity, sway velocity and yaw angular velocity in the motion coordinate system; $\eta = [x, y, \psi]^T$ is the vector composed of X-direction position, Y-direction position and heading angle in the inertial coordinate system; $M = \text{diag}(M_x, M_y, M_\psi)$ is the inertia matrix; $C(v)$ is the centripetal force and Coriolis force matrix; $D(v)$ is the damping matrix; $g(\eta)$ is the restoring force vector; $\tau = [F_u, F_v, F_r]^T$ is the force and moment vector acting on three degrees of freedom.

By combining kinematics and dynamics equations, the model of the underwater vehicle's system can be established as:

$$\dot{x} = \begin{bmatrix} J(\eta)v \\ M^{-1}(F - Cv - Dv - g) \end{bmatrix} = f(x, F), \tag{4}$$

where $x = [x, y, \psi, u, v, r]^T$, and $F = [F_u, F_v, F_r]^T$.

The system behavior in MPC needs to be described by a predictive model. However, for underwater vehicles, the randomness of ocean current direction and velocity have a great impact on its dynamic characteristics. Coupled with the strong nonlinearity and strong coupling of the underwater vehicle's system, it is difficult to establish an accurate system model, which affects the trajectory tracking performance. Therefore, in this paper, the RBFNN identification method is used to identify the dynamic model of underwater vehicles. On the one hand, it can improve the accuracy of model identification; on the other hand, it can modify the network parameters online and suppress the external interference and environmental changes to achieve the purpose of adaptive control.

The trajectory tracking control process of the underwater vehicles studied in this paper is shown in Figure 1.

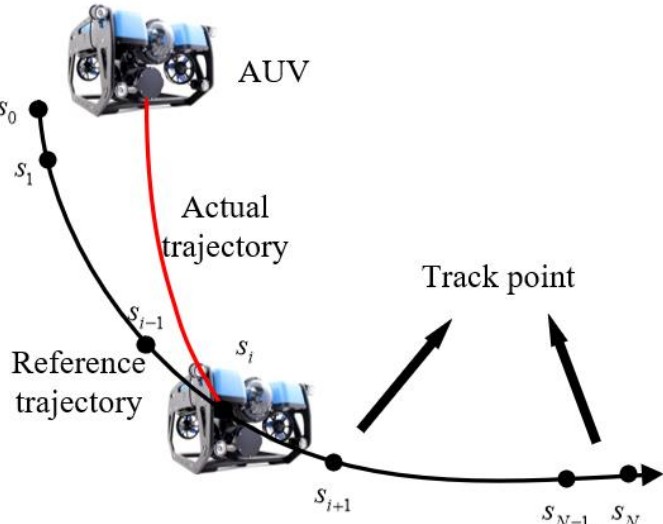

**Figure 1.** The trajectory tracking process diagram of underwater vehicle.

As shown in Figure 1, the reference trajectory $s_1, s_2, \cdots s_N$ defined in inertial coordinate system is composed of $N$ discrete trajectory points from a given initial state. Suppose the trajectory point at a certain time is $s_t = [x_R(t), y_R(t)]^T$. For trajectory tracking control, the reference trajectory should satisfy the physical characteristics constraints and the kinematics equation of the underwater vehicles [19]. That is:

$$
\begin{aligned}
\dot{x}_R &= u_R \cos \psi_R - v_R \sin \psi_R \\
\dot{y}_R &= u_R \sin \psi_R + v_R \cos \\
\dot{\psi}_R &= r_R
\end{aligned}
\tag{5}
$$

According to (5), the real-time state of the underwater vehicle is obtained as:

$$
\begin{cases}
\psi_R = \text{atan2}(\dot{y}_R, \dot{x}_R) \\
u_R = \sqrt{\dot{x}_R^2 + \dot{y}_R^2} \\
v_R = 0 \\
r_R = (\dot{x}_R \ddot{y}_R - \dot{y}_R \ddot{x}_R)/(\dot{x}_R^2 + \dot{y}_R^2)
\end{cases}
\tag{6}
$$

In this paper, a radial basis function neural- nonlinear model predictive control (RBF-NMPC) method is designed to give the corresponding control law at each sampling time, so that the real state $[x, y, \psi, u, v, r]^T$ of the underwater vehicles coincides with the reference trajectory $[x_R, y_R, \psi_R, u_R, v_R, r_R]^T$. The external disturbance will change the state of the underwater vehicle. Therefore, the state information of an underwater vehicle contains the information of external interference. The model accuracy of an underwater vehicle can be improved by adjusting the network parameters. The adaptive ability of the model enables the controller to send out correct control law to ensure that the running state of the underwater vehicles is still on the reference trajectory when there are external disturbances.

## 3. Controller Design

The principle of RBF-NMPC constructed in this paper is shown in Figure 2. The specific implementation process is as follows:

First, from the input and output data, the dynamic model of underwater vehicles is identified off-line by using the IALM-ESC algorithm, so that the RBFNN can basically grasp the dynamic of the underwater vehicle.

Second, in the real-time control stage, in order to improve the accuracy of the NMPC, the error between the system output and the network prediction output is used to adjust

the RBFNN parameters. The adjusted network is used to predict the state quantity in the future.

Finally, in the NMPC optimization stage, the objective function is optimized by the proposed AGWO algorithm, and the control sequence is obtained.

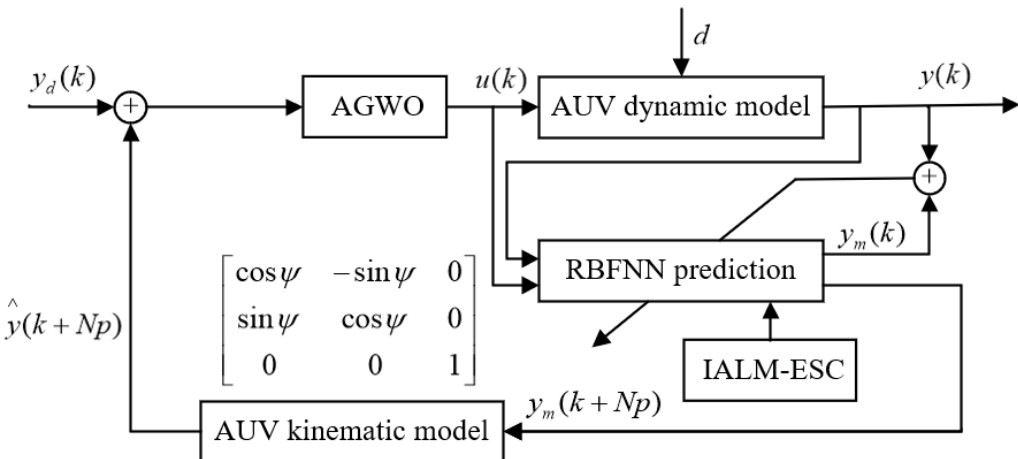

**Figure 2.** Control schematic diagram of the proposed RBF-NMPC.

### 3.1. RBFNN Training

RBFNN is a kind of local approximation neural network with simple structure. By introducing the idea of Gaussian kernel function, any nonlinear system can be approximated with a compact set and any precision. Suppose that there are $r$ neurons in the hidden layer and $x = [x_1, x_2, \cdots x_m]^T$ is the input of the RBFNN, then the output can be expressed as:

$$y = \sum_{j=1}^{r} \omega_j \exp(-\frac{||x - c_j||^2}{2\sigma_j^2}) \tag{7}$$

where $c_j = [c_{j1}, c_{j2}, \cdots c_{jm}]$ is the vector value of the center point of the $j^{th}$ hidden layer neuron, $\sigma_j$ is the width vector of the Gaussian kernel function of the $j^{th}$ hidden layer neuron, and $\omega_j$ is the weight of the $j^{th}$ hidden layer neuron.

When constructing RBFNN, there are two problems to be solved: one is structure identification, the other is parameter estimation. Structure identification is to determine the number of nodes in the hidden layer of the network, and the parameter estimation is to find a set of network parameters (center, band width and weight), which minimums the sample error function (root mean square error):

$$\min E = \sqrt{\frac{1}{n} \sum_{p=1}^{n} (y(p) - y^{ref}(p))^2} \tag{8}$$

where $p$ is the current sample, $n$ is the total number of samples, $y^{ref}$ is the expected output of the sample, and $y$ is the output of RBFNN.

The IALM-ESC applied in this paper is an incremental network construction algorithm. Starting with zero network nodes, the maximum of the error surface is compensated by adding network nodes at the peak or valley of the error surface. The parameters of each new network node are adjusted by the following update rules [20]:

$$\Theta(t+1) = \Theta(t) - (\Psi(t) + \eta(t)I)^{-1}\Omega(t) \tag{9}$$

where $\Psi(t)$ is a quasi-Hessian matrix, $\Omega(t)$ is the gradient vector, $\eta(t)$ is the adaptive damping coefficient, and its adjustment rule is as:

$$\eta(t) = \beta||e(t)|| \tag{10}$$

where $\beta$ is the constant. $\Psi(t)$ and $\Omega(t)$ are the sum of sub matrix $\psi_p(t)$ and sub vector $\omega_p(t)$ of all samples, respectively, and there is:

$$\Psi(t) = \sum_{p=1}^{n} \psi_p(t)$$
$$\Omega(t) = \sum_{p=1}^{n} \omega_p(t) \tag{11}$$

where,

$$\psi_p(t) = j_p^T(t) j_p(t)$$
$$\omega_p(t) = j_p^T(t) e_p(t) \tag{12}$$

where, $j_p(t)$ is the row vector of Jacobian matrix, which is described as:

$$J = \begin{bmatrix} \frac{\partial e_1}{\partial w_1}, & \cdots & \frac{\partial e_1}{\partial w_r}, & \frac{\partial e_1}{\partial c_{11}}, & \cdots & \frac{\partial e_1}{\partial c_{nr}}, & \frac{\partial e_1}{\partial \sigma_1}, & \cdots & \frac{\partial e_1}{\partial \sigma_r} \\ \frac{\partial e_2}{\partial w_1}, & \cdots & \frac{\partial e_2}{\partial w_r}, & \frac{\partial e_2}{\partial c_{11}}, & \cdots & \frac{\partial e_2}{\partial c_{nr}}, & \frac{\partial e_2}{\partial \sigma_1}, & \cdots & \frac{\partial e_2}{\partial \sigma_r} \\ \cdots & & \cdots & \cdots & & \cdots & \cdots & & \cdots \\ \frac{\partial e_p}{\partial w_1}, & \cdots & \frac{\partial e_p}{\partial w_r}, & \frac{\partial e_p}{\partial c_{11}}, & \cdots & \frac{\partial e_p}{\partial c_{nr}}, & \frac{\partial e_p}{\partial \sigma_1}, & \cdots & \frac{\partial e_p}{\partial \sigma_r} \end{bmatrix} \tag{13}$$

According to the update rule of the gradient descent learning algorithm, the elements of row vector of Jacobian matrix can be expressed as:

$$\frac{\partial e_p}{\partial w_j} = \frac{\partial e_p}{\partial y_p} \frac{\partial y_p}{\partial w_j} = -h_j$$
$$\frac{\partial e_p}{\partial c_{ij}} = \frac{\partial e_p}{\partial y_p} \frac{\partial y_p}{\partial h_j} \frac{\partial h_j}{\partial c_{ij}} = -\frac{w_j h_j (x - c_{ij})}{\sigma_j^2}$$
$$\frac{\partial e_p}{\partial \sigma_j} = \frac{\partial e_p}{\partial y_p} \frac{\partial y_p}{\partial h_j} \frac{\partial h_j}{\partial \sigma_j} = -\frac{w_j h_j ||x - c_{ij}||^2}{\sigma_j^3} \tag{14}$$

After adjusting all the parameters, if the root mean square error between the predicted value and the actual value of the RBFNN does not reach the target value, the network nodes will continue to be added at the maximum error, and the node parameters will be trained until the target value is met.

In the off-line phase, the model of the underwater vehicle is initially established. However, in the real-time control stage, the prediction model should also have the ability of adjustment to adapt to the unknown underwater environment. The underwater interference is decomposed into an interference force on each degree of freedom, which causes the state of the underwater vehicle to change. Therefore, it can be considered that the state information of an underwater vehicle contains the information of external interference. According to the error between the system output and the predicted state of the model, the model accuracy of an underwater vehicle can be improved by adjusting. This idea is widely used in the existing adaptive neural network controller of underwater vehicle [21,22]. In this paper, the adaptive gradient descent is used to adjust the network parameter in the real-time control phase to minimize the error between the MPC model and the system output. In each sampling period, after the new sample data are collected, the network parameters are updated once by (15). With the increase in running time, network parameters will be gradually adjusted to adapt to the changes of the external environment.

$$\Theta(t+1) = \Theta(t) - \eta(t) g(t) \tag{15}$$

where $\eta(t)$ is adaptive learning rate, that equal to the adaptive damping coefficient in (10). $g(t)$ is the gradient vector, that composed of factors in (14).

### 3.2. Objective Function and Constraints

The velocity and angular velocity of each degree of freedom of underwater vehicle in the future can be estimated according to the established RBFNN prediction model. The expression is as follows:

$$y_m(k+p|k) = \sum_{j=1}^{r} w_j \exp(-\frac{||x-c_j||^2}{2\sigma_j^2}) \quad p = 1, 2 \cdots N_p \tag{16}$$

where $N_p$ is the prediction horizon, $y_m(k+p|k)$ is the prediction output at the sampling time $k$. $x = [y(k+p-1), \cdots y(k+p-n_A), u(k+p-1), \cdots u(k+p-n_B)]^T$ is the input, and it will change with the prediction horizon. $n_A$ and $n_B$ are the order of the output and input, respectively. Then the position and attitude $[x, y, \psi]^T$ of the underwater vehicle in the prediction horizon are calculated by the kinematics equation, and all the state variables $\hat{y}(k+p|k)$ of the underwater vehicle in the prediction horizon are obtained.

In this paper, the minimum value of quadratic objective function is used to express the optimization performance index at $k$ time [23]. The expression is as follows:

$$\begin{aligned} \min_{\Delta u(k)} &\left\{ Q \sum_{p=1}^{N_p} (\widetilde{y}(k+p|k))^2 + \lambda \sum_{p=0}^{N_u-1} (\Delta u(k+p|k))^2 \right\} \\ s.t. \quad &u^{\min} \leq u(k+p|k) \leq u^{\max} p = 0, 1 \cdots, N_u - 1 \\ &-\Delta u^{\max} \leq \Delta u(k+p|k) \leq \Delta u^{\max} p = 0, 1 \cdots, N_u - 1 \\ &y^{\min} \leq \hat{y}(k+p|k) \leq y^{\max} p = 0, 1 \cdots, N_p \end{aligned} \tag{17}$$

where $\widetilde{y}(k+p|k) = y^{sp}(k+p|k) - \hat{y}(k+p|k)$ is the difference between reference trajectory and model prediction. $\Delta u(k+p|k) = u(k+p|k) - u(k+p-1|k)$ is the control increment. $N_p$ and $N_u$ represent the prediction horizon and control horizon. $Q$ and $\lambda$ are corresponding weighting matrices. $u^{\min}$ and $u^{\max}$ are the upper and lower bounds of $u(k+p|k)$. $-\Delta u^{\max}$ and $\Delta u^{\max}$ are the upper and lower bounds of $\Delta u(k+p|k)$. $y^{\min}$ and $y^{\max}$ are the upper and lower bounds of $\hat{y}(k+p|k)$, respectively.

### 3.3. AGWO Algorithm

For solving NMPC, the traditional gradient descent method has limitations in calculating capability [24]. Biological heuristic optimization algorithm has been proved to have strong application potential in complex NMPC problems [25]. In [26], a modified GWO and the Moth-Flame Optimization were proposed to improve the performance when applied as an NMPC solver. The GWO algorithm simulates the predatory behavior of gray wolf group and achieves the goal of optimization based on the mechanism of wolf group cooperation [27]. In GWO algorithm, the first three wolves with the best fitness (optimal solution: $\alpha$, $\beta$ and $\delta$) guide other wolves to search for the target. The remaining wolves (candidate solutions) are defined as $\omega$, and they update their positions around $\alpha$, $\beta$ and $\delta$. The distance between the individual and the prey is shown in (18), and its position update is shown in (19).

$$\vec{D}_p(t) = |\vec{C} \cdot \vec{X}_P(t) - \vec{X}(t)| \tag{18}$$

$$\vec{X}_{D_P}(t) = \vec{X}_p(t) - \vec{A} \cdot \vec{D}_p(t) \quad P = \alpha, \beta, \delta \tag{19}$$

where $t$ is the current iteration times. $\vec{A}$ and $\vec{C}$ are the coefficient vectors. $\vec{X}_p$ and $\vec{X}_{D_P}$ are the position vectors of prey and gray wolf, respectively, and there is $\vec{A} = 2\vec{a} \cdot \vec{r}_1 - \vec{a}$, $\vec{C} = 2 \cdot \vec{r}_2$. $\vec{a}$ is the convergence factor and decreases linearly from 2 to 0 with the number of iterations. The modules of $\vec{r}_1$ and $\vec{r}_2$ are random numbers between [0, 1].

The location update of each search factor is expressed as:

$$X(t+1) = \frac{\vec{X}_{D_\alpha}(t) + \vec{X}_{D_\beta}(t) + \vec{X}_{D_\delta}(t)}{3} \tag{20}$$

With the increase in the number of iterations $t$, the GWO algorithm finally finds the optimal solution through the way of trapping. Although the GWO algorithm has been widely used, it also has the characteristics of slow convergence speed and is easy to be limited to local minimum [28]. In order to improve the performance of the GWO algorithm, an adaptive strategy is applied in this paper. The adaptive weight is added to Equation (20) to speed up the convergence speed. In addition, the ability to jump out of the local optimum can be improved by crossing the worst set of each iteration.

In general, wolf $\alpha$ has the highest command in the pack. However, in some special cases, $\beta$ and $\delta$ can also command wolves temporarily. Therefore, the position of the wolf pack must be iterated according to different weights. However, if the weight is fixed, it will not be conducive to the regeneration of the population. Therefore, an adaptive weighted position updating method is applied, as shown in (21).

$$X(t+1) = \frac{\lambda_\alpha(t+1)\vec{X}_{D_\alpha}(t) + \lambda_\beta(t+1)\vec{X}_{D_\beta}(t) + \lambda_\delta(t+1)\vec{X}_{D_\delta}(t)}{3} \tag{21}$$

where $\lambda_\alpha$, $\lambda_\beta$ and $\lambda_\delta$ are the adaptive weights in each iteration, which can be expressed as:

$$\lambda_p(t+1) = M_{\lambda_p}(t+1) + 0.1 \cdot \text{randn}(0,1), \quad P = \alpha, \beta, \delta \tag{22}$$

where, $\text{randn}(0,1)$ is the standard normal distribution; $M_{\lambda_p}$ is the average value of weight update, which can be expressed as:

$$M_{\lambda_p}(t+1) = (1-c) \cdot M_{\lambda_p}(t) + c \cdot \text{mean}(G_{\lambda_p}), \quad P = \alpha, \beta, \delta \tag{23}$$

where $c$ is a constant, which is set to 0.1 in this paper. $G_{\lambda_p}$ is a file which stores better weight than the last iteration. In the initialization phase, the stored weights $G_{\lambda_p}$ are all 1.

In addition, inspired by the differential evolution algorithm, the idea of worst-case crossover is proposed in order to improve the global search performance of the GWO algorithm. The search factor set with poor fitness in each iteration exchanges information with $\alpha$, $\beta$ and $\delta$, so as to increase the diversity of the population. The ability to jump out of the local optimum can be improved by crossing the population. Set the number of bad sets as K. The information exchange formula of each difference factor in K is expressed as follows:

$$x_{least,d}(t+1) = \begin{cases} x_{\alpha,d}(t) & if \ a \leq 1 \\ x_{\beta,d}(t) & if \ 1 < a \leq 2 \\ x_{\delta,d}(t) & if \ 2 < a \leq 3 \end{cases} \tag{24}$$

where $a$ is a random number between [0, 3], and $d$ is the dimension of search factor.

Through the above improvements, the proposed AGWO algorithm has a great improvement in the convergence speed and the search ability of the global optimal value when compared with the traditional GWO algorithm.

The AGWO algorithm proposed in this paper is applied to the optimization problem shown in (17), and the first control sequence of the optimization result is taken as the optimal control law. Then it is loaded into the underwater vehicle's system for real-time control. The flow chart of the proposed RBF-NMPC Algorithm 1 is as follows:

| **Algorithm 1** RBF-NMPC | |
|---|---|
| 1: | Develop RBFNN predictive model offline using offline data; |
| 2: | Initialize the parameters of RBF-NMPC; |
| 3: | **For** k = 1 to N **do** |
| 4: | Sample the plant output $y(k)$; |
| 5: | Update the parameters of RBFNN to adapt the real environment; |
| 6: | Calculate the prediction outputs $\hat{y}(k + p)$; |
| 7: | **While** current iteration times $t < t_{max}$; |
| 8: | Compute the control signal by AGWO; |
| 9: | $t + +$; |
| 10: | **End while** |
| 11: | Sent the control signal to the underwater vehicle; |
| 12: | **end for** |

## 4. Simulation Results

### 4.1. Model Identification Results

In order to verify the effectiveness of the proposed model identification based on RBFNN, the random step signals are used as the excitation signal to obtain the state response of the underwater vehicle, which are taken as the model training sample. The width of each step signal represents the excitation action time, which reflects the relationship between the dynamic response of the underwater vehicle and the excitation frequency. The dynamic information of the underwater vehicle can be more captured by random step signal. According to the thrust constraints of each propeller, the thrust moment range of $F_u$ is set as $[-2000 \text{ N}, 2000 \text{ N}]$, the thrust range of $F_v$ is set as $[-2000 \text{ N}, 2000 \text{ N}]$, and the range of $F_r$ is set as $[-900 \text{ Nm}, 900 \text{ Nm}]$. The input of the RBFNN is $x(k) = [F_u(k-1), F_v(k-1), F_r(k-1), u(k-1), v(k-1), r(k-1)]$, and the actual output of the RBFNN is $y(k) = [u(k), v(k), r(k)]$. Figure 3 shows 2000 sets of input and output data for underwater vehicles dynamic model identification, where 1900 groups are used as training data and 100 groups are used as test data.

Before the neural networks training based on the IALM-ECS, the data preprocessing is carried out to prevent system instability or slow training speed caused by different dimensions. The target mean square error is set at 0.05. In Figure 4, the root mean square error (RMSE) of the network output decreases with the increase in the number of nodes. After reaching the target value, the number of network nodes stops increasing. Based on the IALM-ECS algorithm, RBFNN is constructed incrementally from zero nodes, which makes the network more compact and has good generalization ability. Compared with the traditional trial and error method, there is no randomness in the whole process.

The root mean square error curve of RBFNN offline training is shown in Figure 5. It can be seen that the proposed IALM-ESC algorithm can improve the convergence speed. Even if the initialization error is large, it can converge about 100 iterations. However, the traditional GD method cannot get the better parameters when it reaches 500 iterations.

The comparison between the actual and predicted values of the test set is shown in Figure 6. It can be seen that the actual value is basically consistent with the predicted value. This shows that the network structure is simple, and the modeling effect is satisfactory.

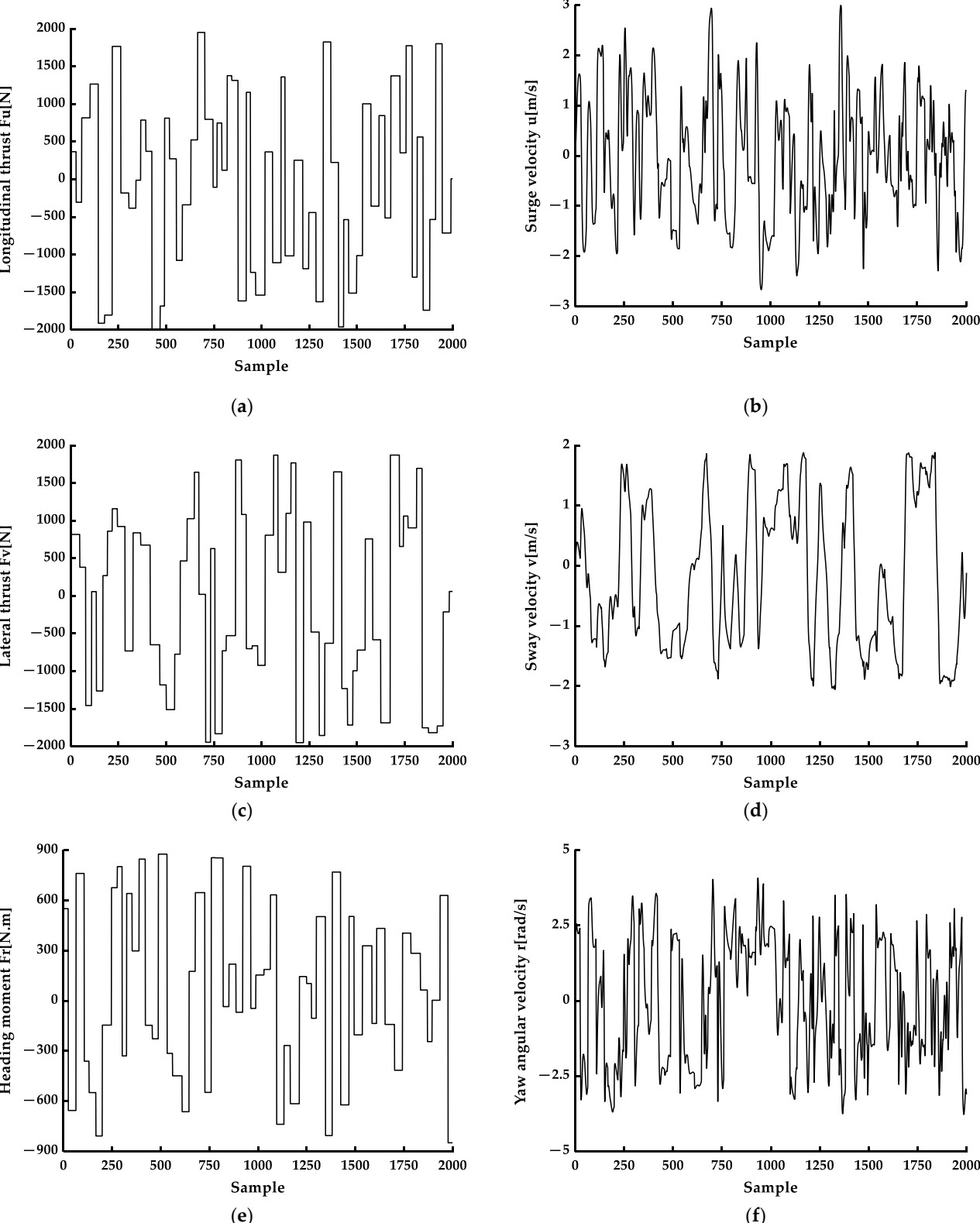

**Figure 3.** The sets of input and output data for dynamic system identification: (**a**) The excitation signal of longitudinal thrust; (**b**) The response signal of surge velocity; (**c**) The excitation signal of lateral thrust; (**d**) The response signal of sway velocity;(**e**) The excitation signal of heading moment; (**f**) The response signal of yaw angular velocity.

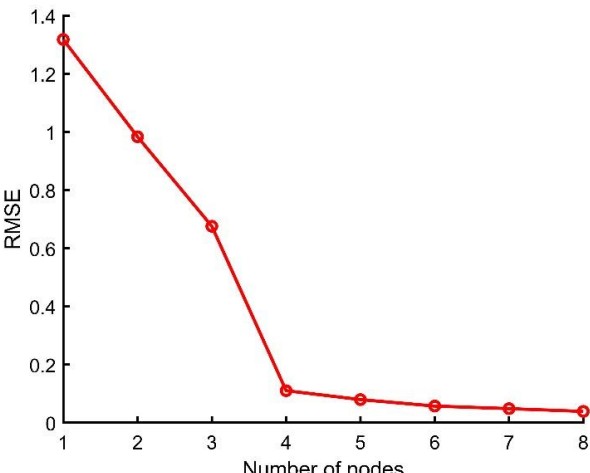

**Figure 4.** The structure identification diagram of IALM-ESC algorithm.

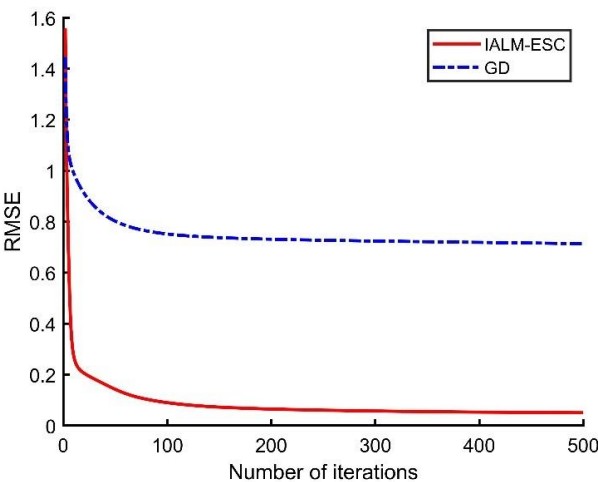

**Figure 5.** Convergence graph of RBFNN parameter learning training based on IAMM-ESC and GD.

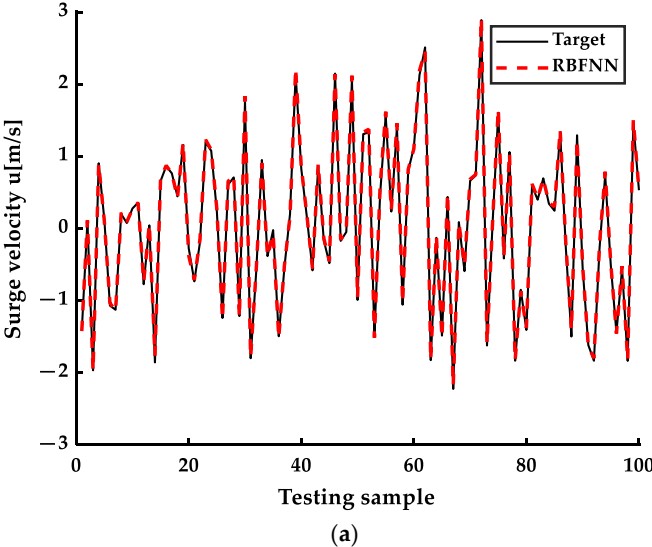

(**a**)

**Figure 6.** *Cont*.

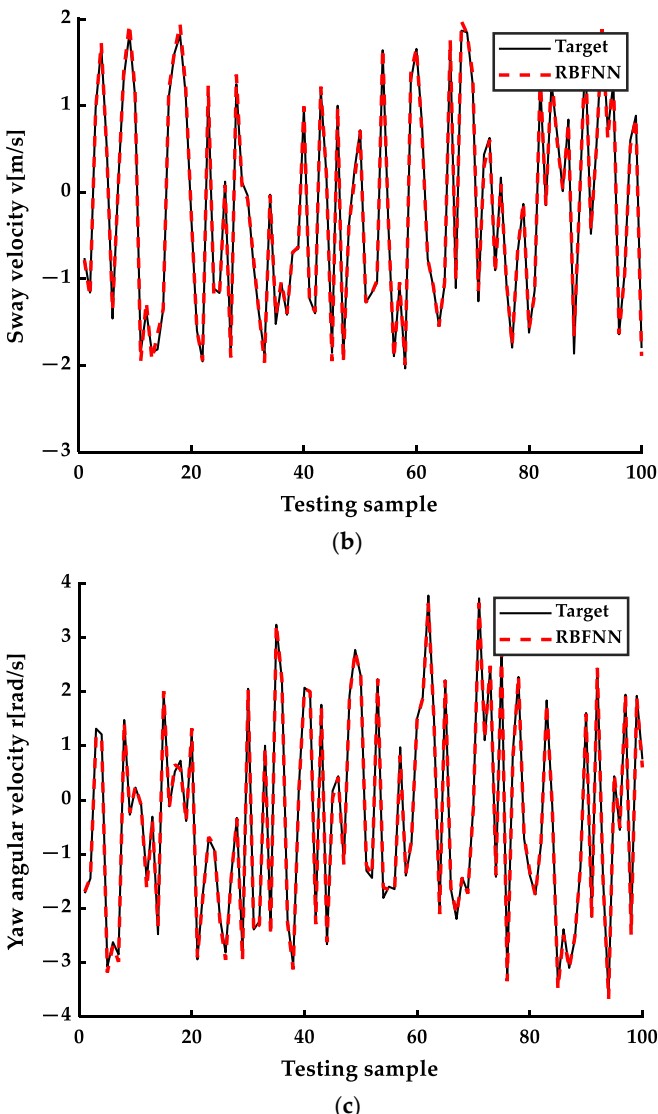

**Figure 6.** Comparison chart of actual value and predicted value of test set: (**a**) The actual and predicted values of surge velocity; (**b**) The actual and predicted values of sway velocity; (**c**) The actual and predicted values of yaw angular velocity.

*4.2. Optimization Results of AGWO*

Two typical functions (convex function and nonconvex function) are chosen to test the optimization performance of AGWO, and the results are shown in Figures 7 and 8. First of all, the two figures show that the optimization effect of AGWO is better than the other three methods. It can be seen from Figure 7 that AGWO has a great improvement in the convergence speed compared with GWO, especially in the later stage. It indicates that the existence of adaptive weights can accelerate the convergence of AGWO. In Figure 8, when particle swarm optimization (PSO), differential evolution (DE) and GWO all fall into the local optimal value, AGWO can jump out of the local optimal value and achieve a good optimization result. The results show that the AGWO proposed in this paper has a great improvement in convergence speed and convergence effect.

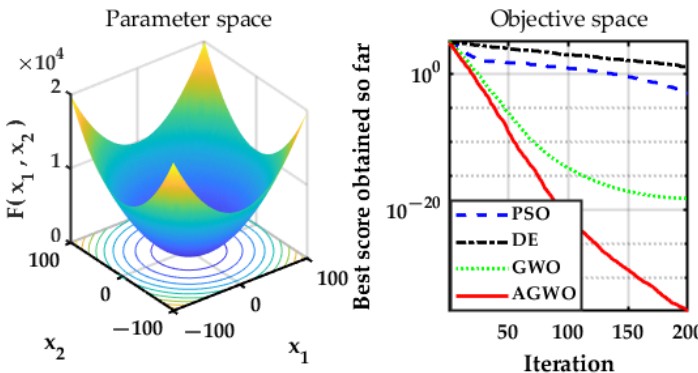

**Figure 7.** Convergence effect of convex function.

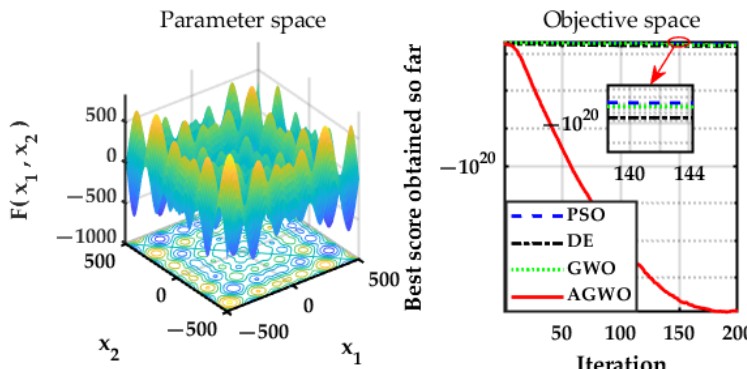

**Figure 8.** Convergence effect of nonconvex function.

### 4.3. Trajectory Tracking Control Results

In order to verify the effectiveness of the proposed trajectory tracking control method, the reference trajectory is selected as follows:

$$p(t) = \begin{cases} x_R = t \\ y_R = \sin t \end{cases} \tag{25}$$

In the simulation, the sampling period is $\Delta t = 0.05$ s. The prediction horizon is $N = 5\Delta t$. The control horizon is $N_u = 2\Delta t$. The state quantity weighting coefficient is $Q = diag(10^4, 10^4, 10^2, 10^1, 10^1, 10^1)$. The control quantity weighting coefficient is $\lambda = diag(10^{-4}, 10^{-4}, 10^{-4})$.

The tracking results of the two methods (RBF-NMPC and LMPC) are shown in Figure 9. Combining with Figure 9a,b, it can be seen that both the overall tracking and the tracking in each state of RBF-NMPC are highly consistent with the reference trajectory. On the contrary, due to the simplification of the underwater vehicle model in LMPC, there are large modeling errors, and the trajectory tracking control error is larger. The optimization time of the two methods is shown in Figure 10. In a control cycle, the network parameters are adjusted once according to (15), and then the state prediction and control law optimization are carried out. The average time for solving calculation of RBF-NMPC is 0.0144 s and that of LMPC is 0.0076 s (The time is measured with the time function in MATLAB). Although the proposed RBF-NMPC takes longer, the optimization time of the two methods is much lower than the sampling time, which can ensure the real-time tracking. The simple network structure and the convergence effect of AGWO ensure that the optimization time of RBF-NMPC can reach the same level as LMPC.

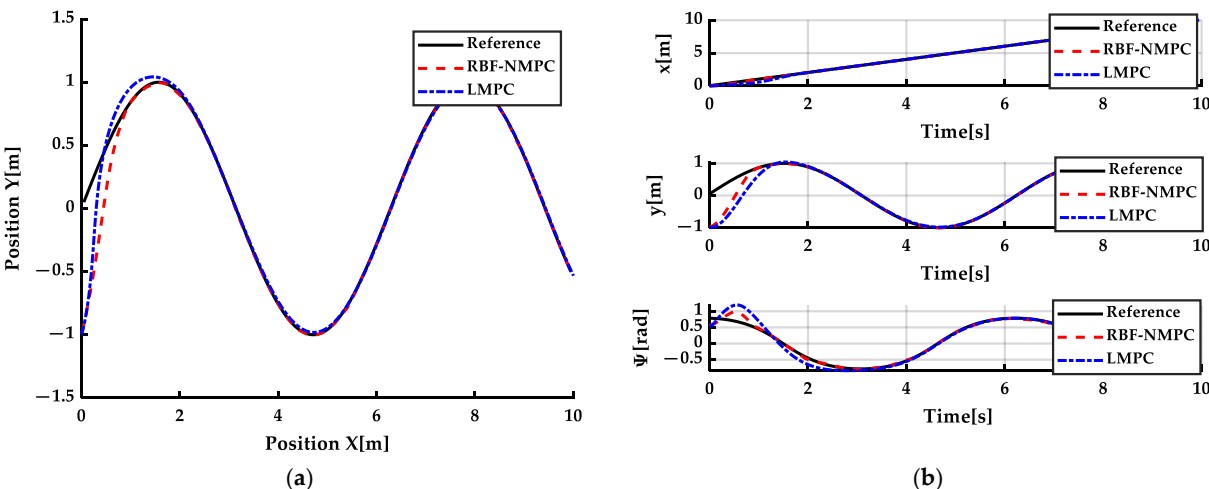

**Figure 9.** Tracking control comparison results: (**a**) Tracking effect in xoy plane; (**b**) Tracking effect of each state.

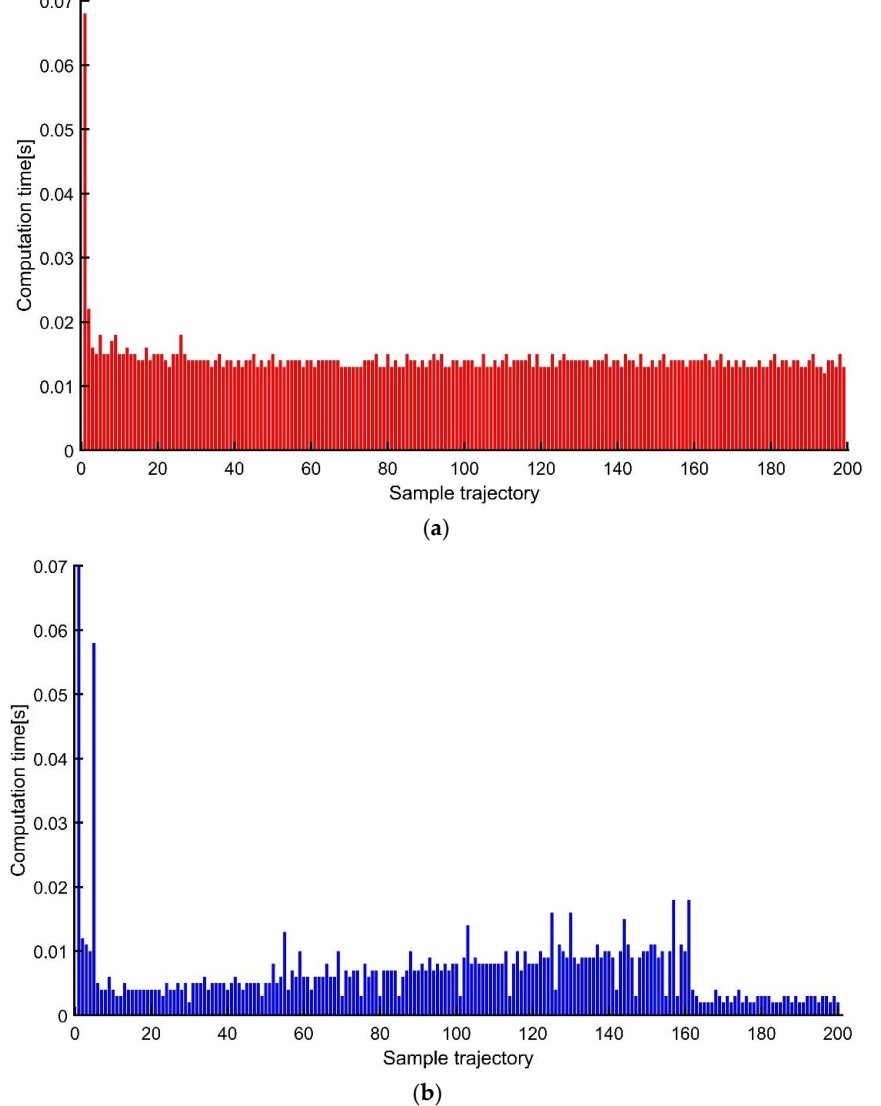

**Figure 10.** Computation time of the RBF-NMPC and LMPC: (**a**) Computation time of the RBF-NMPC; (**b**) Computation time of the LMPC.

In addition, different optimization algorithms are used to solve the same NMPC problem. Based on the obtained control laws, trajectory tracking control is carried out respectively. The mean square error (MSE) for each optimization algorithm in trajectory tracking are summarized in Table 1. The tracking performance of AGWO is significantly improved in each degree of freedom compared with other optimization algorithms.

**Table 1.** The mean square error (MSE) in trajectory tracking for each optimization algorithm.

| MSE | PSO | DE | GWO | AGWO | Improvement (AGWO to GWO) |
|---|---|---|---|---|---|
| $x[m^2]$ | 0.1342 | 0.0026 | 0.0028 | 0.0012 | 57% |
| $y[m^2]$ | 0.0703 | 0.0160 | 0.0158 | 0.0111 | 30% |
| $\psi[rad^2]$ | 0.1312 | 0.0059 | 0.0059 | 0.0047 | 20% |

In order to verify the anti-interference ability of RBF-NMPC, the trajectory tracking control under the interference environment is carried out. In the simulation, unknown random interference is added to each degree of freedom, and the expression is shown in (26). The simulation results under the interference environment are shown in Figures 11 and 12.

$$D = \begin{cases} 400\sin(\frac{k\pi}{100}) + rand(-400, 400) \\ 400\sin(\frac{k\pi}{100}) + rand(-400, 400) \\ 200\sin(\frac{k\pi}{100}) + rand(-200, 200) \end{cases} \tag{26}$$

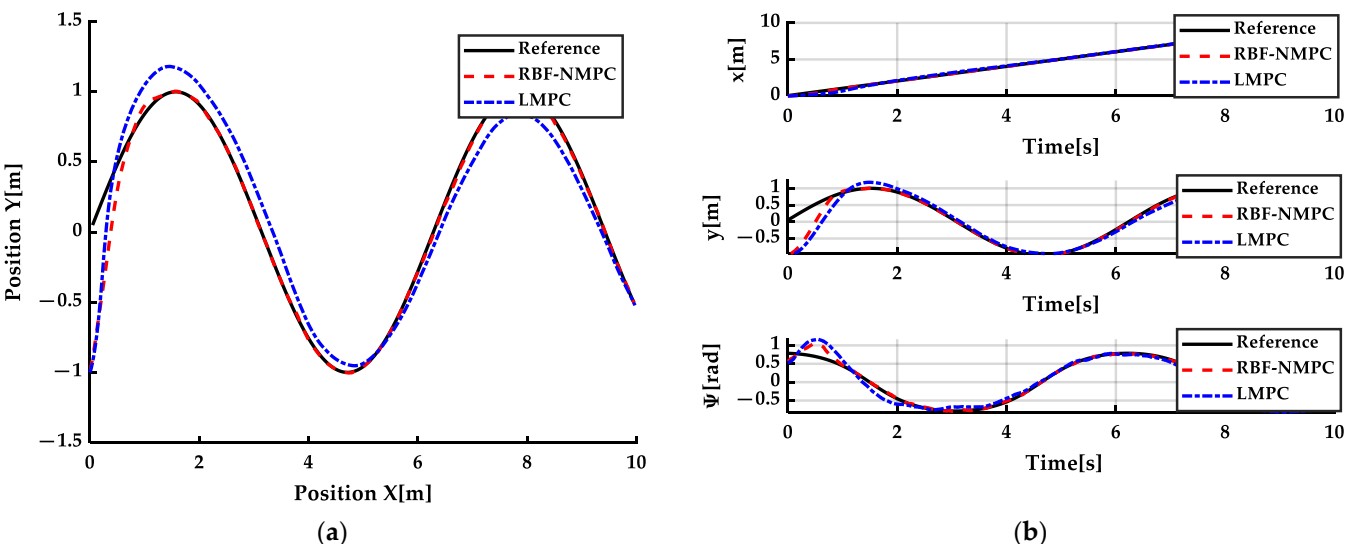

(a)

(b)

**Figure 11.** Tracking control comparison results under interference environment: (**a**) Tracking effect in xoy plane; (**b**) Tracking effect of each state.

It can be seen from Figure 11 that LMPC cannot track the reference trajectory well under interference environment, especially since the deviation between the heading angle and reference value is large. In contrast, RBF-NMPC can update the model according to the real value of the system model output after each optimization, which shows superior adaptive ability in a complex working environment, and its trajectory tracking control effect is better. Figure 12 shows the optimization times of the two methods. Under interference environment, the average time for solving calculation of RBF-NMPC is 0.0146 s, and that of LMPC is 0.0083 s. The optimization time of RBF-NMPC is still far less than the sampling time. At the same time, RBF-NMPC has the ability of adaptive parameter adjustment, which will show excellent tracking performance in complex conditions.

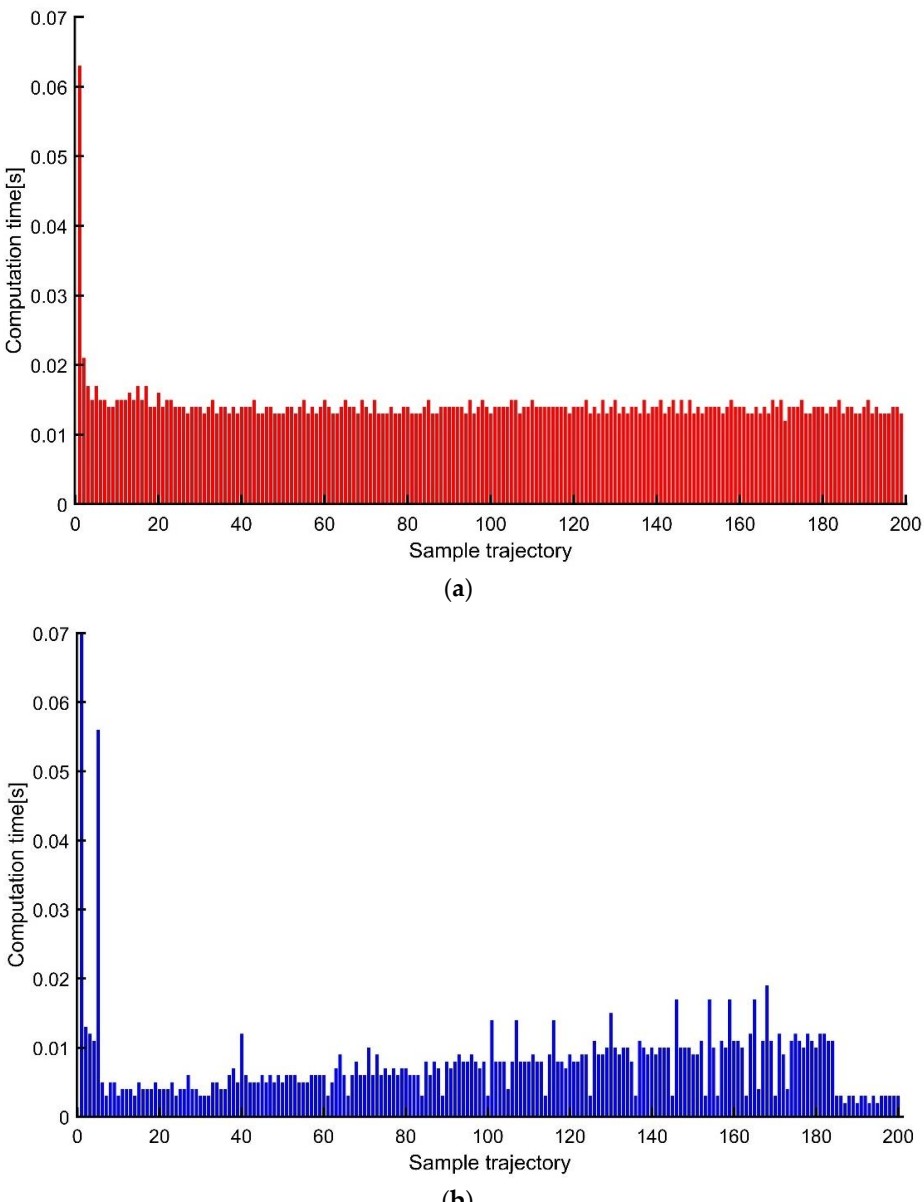

**Figure 12.** Computation time of the RBF-NMPC and LMPC under interference environment: (**a**) Computation time of the RBF-NMPC; (**b**) Computation time of the LMPC.

The mean square errors for trajectory tracking are summarized in Table 2. The tracking performance of RBF-NMPC is improved by at least 43% and 25% in the case of no interference and interference, respectively.

**Table 2.** The mean square error (MSE) of the RBF-NMPC and LMPC in trajectory tracking.

| Environment | MSE | LMPC | RBF-NMPC | Improvement |
|---|---|---|---|---|
| No interference | $x[m^2]$ | 0.0095 | 0.0012 | 87% |
| | $y[m^2]$ | 0.0196 | 0.0111 | 43% |
| | $\psi[rad^2]$ | 0.0096 | 0.0047 | 51% |
| Interference | $x[m^2]$ | 0.0106 | 0.0026 | 75% |
| | $y[m^2]$ | 0.0205 | 0.0154 | 25% |
| | $\psi[rad^2]$ | 0.0115 | 0.0051 | 56% |

## 5. Conclusions

In this paper, an adaptive RBF-NMPC trajectory tracking control algorithm is proposed for underwater vehicles. This method combines the NMPC with RBFNN and AGWO algorithm. It solves the problems of modeling difficulty and poor real-time performance in the application of NMPC in underwater vehicles. Simulation results show that the trajectory tracking performance of RBF-NMPC is greatly improved compared with the LMPC and traditional optimization algorithms. In the near future, how to reduce the time of first optimization, and how to combine RBF-NMPC with robust control to make the whole trajectory tracking control system more stable, will be our next research direction.

**Author Contributions:** Conceptualization, Z.C.; methodology, D.W.; software, D.W.; writing—original draft preparation, D.W.; writing—review and editing, Z.C. and F.M. All authors have read and agreed to the published version of the manuscript.

**Funding:** This research was funded by the National Natural Science Foundation of China under Grant U2006228, 51839004, and in part by the Shanghai Science and Technology Innovation Action Plan under Grant 18550720100, 19040501600.

**Institutional Review Board Statement:** Not applicable.

**Informed Consent Statement:** Not applicable.

**Data Availability Statement:** Not applicable.

**Conflicts of Interest:** The authors declare no conflict of interest.

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
