# Peer review of "An Adaptive RBF-NMPC Architecture for Trajectory Tracking Control of Underwater Vehicles"

_machines, doi:10.3390/machines9050105_

Round 1

Reviewer 1 Report

  • Interesting research.
  • Well worded claims (e.g. discussion versus development of RBFNN and NMPC) lend credibility to the authors proposal of an instantiation of AGWO.
  • Figures are well presented.
    • Figure 2 has internal text that will become difficult to discern when printed. Please maintain text size inside figures no smaller than the figure caption (the smallest text size permissible in the template) to ensure legibility.
    • Definitions in the figure captions of variables inside figures will increase readability of the manuscript
    • Abscissa and ordinate scales are far too small in figure 3,10  and 12 and render the (sub)figures largely useless to the reader.
    • Identical line styles and sizes were used in figure 5, 7, 8, and 9 making the data indistinguishable when printed (particularly in black and white). Please modify one of the line styles or thickness to insure legibility.
    • Identical line sizes were used in figure 6 making the dashed data indistinguishable from the solid-lined data when printed (particularly in black and white). Please modify one of the line thickness to insure legibility.
    • Figure 11 has three weaknesses, blurriness, identical line styles and thicknesses, and illegible text font of abscissa and ordinate.
  • Presentation of the algorithm is well done in a pithy flow chart
  • Simulations are modestly well presented, but are not supported by broadly described figures of merit. Figure 8 could provide meaningful quantitative comparison by simply providing the means and standard deviations of the tracking errors in a table. Percent improvement over a declared benchmark are ubiquitously understandable results in broad terms, and such terms belong in the manuscript’s abstract and conclusions.
  • No future research was recommended permitting the reader to consider taking the next steps in the lineage of research. The reviewer recommends comparison of the proposed AGWO algorithm to the recently published deterministic artificial intelligence proposed for trajectory tracking of unmanned underwater vehicles.

Author Response

Machines

AUTHOR REPLY TO REVIEWER #1

We greatly appreciate the anonymous reviewer’s comments and contributions to this paper.

1. Well worded claims (e.g. discussion versus development of RBFNN and NMPC) lend credibility to the authors proposal of an instantiation of AGWO.

Answer: Thank you very much. In the revised manuscript, the content about development of RBFNN and NMPC and some relevant references have been added to lend credibility to the proposal of AGWO. (to see the line 59-62 and line 228-230)

  1. Figures are well presented.

2.1 Figure 2 has internal text that will become difficult to discern when printed. Please maintain text size inside figures no smaller than the figure caption (the smallest text size permissible in the template) to ensure legibility.

Answer: Thank you for your advice. The text size inside the figure 2 has been increased. (to see the figure 2)

2.2 Definitions in the figure captions of variables inside figures will increase readability of the manuscript.

Answer: Thank you for your advice. The definitions in the figure captions of variables inside figures has been added. (to see the figure 3,6,9,11)

2.3 Abscissa and ordinate scales are far too small in figure 3,10 and 12 and render the (sub)figures largely useless to the reader.

Answer: Thank you for your advice. We have increased the abscissa and ordinate scales in figure 3,10 and 12 to ensure that the information in the figure can be clearly presented to the readers (to see the title figure 3,10,12)

2.4 Identical line styles and sizes were used in figure 5, 7, 8, and 9 making the data indistinguishable when printed (particularly in black and white). Please modify one of the line styles or thickness to insure legibility.

Answer: Thank you for your advice. One of the line styles has been modified to insure legibility in figure 5,7,8,9. (to see the figure 5,7,8,9)

2.5 Identical line sizes were used in figure 6 making the dashed data indistinguishable from the solid-lined data when printed (particularly in black and white). Please modify one of the line thickness to insure legibility.

Answer: Thank you for your advice. One of the line thickness has been modified to insure legibility in figure 6. (to see the figure 6)

2.6 Figure 11 has three weaknesses, blurriness, identical line styles and thicknesses, and illegible text font of abscissa and ordinate.

Answer: Thank you very much. Figure 11 has been reformulated to insure the clarity. One of the line style has been modified and the text font of abscissa and ordinate has been modified in figure11. (to see the figure 11)

 3. Presentation of the algorithm is well done in a pithy flow chart

Answer: Thank you for your advice. The flow chart has been modified in a pithy form. (to see the line 274)

4. Simulations are modestly well presented, but are not supported by broadly described figures of merit. Figure 8 could provide meaningful quantitative comparison by simply providing the means and standard deviations of the tracking errors in a table. Percent improvement over a declared benchmark are ubiquitously understandable results in broad terms, and such terms belong in the manuscript’s abstract and conclusions.

Answer: Thank you for your advice. The mean square error in trajectory tracking for each optimization algorithm and the mean square error of the RBF-NMPC and LMPC in trajectory tracking have been shown in Table 1(line 366) and Table 2(line 393). Percent improvement has been added to the abstract. (to see the line 366, line 393 and the abstract)

5. No future research was recommended permitting the reader to consider taking the next steps in the lineage of research. The reviewer recommends comparison of the proposed AGWO algorithm to the recently published deterministic artificial intelligence proposed for trajectory tracking of unmanned underwater vehicles.

Answer: Thank you for your advice. The comparative experiment has been implemented. The comparison of the proposed AGWO algorithm to the other deterministic artificial intelligence proposed for trajectory tracking is shown in Table 1(line 366). The future research has been recommended in the conclusions. (to see the conclusion and line 366)

Reviewer 2 Report

1. There are less conclusion  and loss of some digital results in the abstract.

2. As words in abstract  from line 12- 14, network parameters are adjusted by using error between system output and network prediction, how the predication  can adaptive to the real environment  with no priori real environment sensing data ?

3. In line 42, how can underwater vehicle  be considered as non linear system ?

4. Between line 48-50 , the logic is some obscure. whether online adaptive correction equal to linear system theory ? 

5. NMPC, MPC ,6-DOF , these specs should be clearly express

6. Which induced these kinds of phenomena strong nonlinearity and frequent disturbances in line 63 , that should be discussed .

7. Between line 72 and 72 ,for the  trajectory tracking control architecture , if this kind of control method  that can be called identification like system  identification?  

8. In line 93-96 usually ... this paper studies ... , the logic needs some improvement .

9.  In the paper,  line 258 -259 , the random step signals are used as the excitation signal ,this should have some discuss why choosing this kind of excitation signal , for general consideration , control signal should not be random.

10. For equ (15) ,  variables need be explained clearly.

11.In the line 328-330, the average optimization time RBF-NMPC is 0.0144 s and that of LMPC is 0.0076 s., how to measure and get this time ,as the sample period said as 0.05s ,in the line 319.

12. In the all,  in this paper , there is short in the procedure of training and how to adjust to adapt the real environment . In this part, paper need to supply more solid contains .

13. Over all , English spell and writing style should be improved. 

Author Response

Machines

AUTHOR REPLY TO REVIEWER #2

We greatly appreciate the anonymous reviewer’s comments and contributions to this paper.

1. There are less conclusion and loss of some digital results in the abstract.

Answer: Thank you for your advice. The simulation experiment of quantitative analysis has been implemented in Table 1(line 366) and Table 2(line 393). The conclusion and some digital results have been added to the abstract. (to see the abstract, the line 366 and line 393)

2. As words in abstract from line 12- 14, network parameters are adjusted by using error between system output and network prediction, how the predication can adaptive to the real environment with no priori real environment sensing data?

Answer: Thank you very much. This part has been supplemented in detail in the text. The prediction model should have the ability of adjustment to adapt to the unknown underwater environment. The real state of the underwater vehicle needs to be collected continuously, and the network parameters are corrected by using the error of real state and prediction state. Adaptive gradient descent is used to adjust the online parameter set in the real-time control process to adapt the real environment. (to see the line 195-206)

3. In line 42, how can underwater vehicle be considered as nonlinear system?

Answer: Thank you very much. Underwater vehicle is a typical nonlinear system, which has high coupling, nonlinear and time-varying characteristics. The nonlinear effects of hydrodynamic damping, Coriolis, and centripetal forces will affect the tracking performance when the underwater vehicle is working. (to see the conclusion and line 47-50)

4. Between line 48-50, the logic is some obscure. whether online adaptive correction equal to linear system theory?

Answer: Thank you very much. The relevant content has been rewritten. The online adaptive correction is an advantage of the RBF-NMPC proposed in this paper. The dynamic model of the underwater vehicle established by Newton-Euler equation do not have the function of online adaptive correction, so it is difficult to ensure the accuracy and applicability when the underwater vehicle is in a complex environment. The shortcoming of the current common method is used to propose the RBF-NMPC in this part. (to see the line 53-57)

5. NMPC, MPC ,6-DOF, these specs should be clearly express

Answer: Thank you very much. NMPC, MPC and 6-DOF have been expressed clearly. (to see the line 43, line 50 and line 67)

6. Which induced these kinds of phenomena strong nonlinearity and frequent disturbances in line 63, that should be discussed.

Answer: Thank you very much. The strong nonlinearity of underwater vehicle is induced by complexity and nonlinearity of dynamic model. The frequent disturbance is induced by ocean current. (to see the line 70-72)

  7. Between line 72 and 72, for the trajectory tracking control architecture, if this kind of control method that can be called identification like system identification?

Answer: Your question is very correct, which is also one of the key issues in this paper. The improved adaptive Levenberg Marquardt-error surface compensation (IALM-ESC) algorithm is applied to the underwater vehicle system identification, and fewer network nodes are used to reflect the dynamic characteristics of underwater vehicles. (to see the line 80-82)

8. In line 93-96 usually ... this paper studies ..., the logic needs some improvement.

Answer: Thank you very much. The previous sentence in this part has been deleted to improve the logic. This paper studies the trajectory tracking control of the underwater vehicles in the horizontal plane, and the motion of surge, sway and yaw are considered. (to see the line 103-104)

9. In the paper, line 258 -259, the random step signals are used as the excitation signal, this should have some discuss why choosing this kind of excitation signal, for general consideration, control signal should not be random.

Answer: Thank you very much. The dynamic information of the underwater vehicle can be more captured by random step signal because of the characteristics of high coupling, nonlinear for underwater vehicle. The width of each step signal represents the excitation action time, which reflects the relationship between the dynamic response of the underwater vehicle and the excitation frequency. (to see the line 278-283)

10. For equ(15), variables need be explained clearly.

Answer: Thank you for your advice. variables have been explained clearly in equ(15).(to see the line 210)

11. In the line 328-330, the average optimization time RBF-NMPC is 0.0144 s and that of LMPC is 0.0076 s., how to measure and get this time, as the sample period said as 0.05s, in the line 319.

Answer: Thank you very much. The time is measured by etime function in MATLAB. The optimization time of each step is measured, and an average value is obtained finally. (to see the line 349)

12. In the all, in this paper, there is short in the procedure of training and how to adjust to adapt the real environment. In this part, paper need to supply more solid contains.

Answer: Thank you for your advice. The more solid contains about the procedure of training and the adaption to the real environment have been supplied. The selection of excitation signal is supplemented in the line 278-283. The detail about adaption to the real environment has been supplemented in the line 195-206. (to see the line 278-283 and line 195-206)

13. Over all, English spell and writing style should be improved.

Answer: Thanks for your advice. We invited professionals to polish the paper. At the same time, we confirm the correctness of text content description.

Reviewer 3 Report

In this paper, the author proposed an adaptive control algorithm based on RBF neural network (RBFNN) and nonlinear model predictive control (NMPC) for underwater vehicle.This work has a certain research value, but it has not yet reached the publishing standard. Some suggestions on revision are as follows:

1) The abstract and conclusions of this article need to be revised, and some conclusions of quantitative analysis should be put forward.

2) The language in this manuscript needs to be further improved and polished.

3) The clarity of Figure 5,6,7,8,9,10,11 and 12 are needed to be improved.

In this condition, I suggest minor revisions to this manuscript.

Author Response

Machines

AUTHOR REPLY TO REVIEWER #3

We greatly appreciate the anonymous reviewer’s comments and contributions to this paper.

1. The abstract and conclusions of this article need to be revised, and some conclusions of quantitative analysis should be put forward.

Answer: Thank you very much. In the revised manuscript, the abstract and conclusions have been modified. Some conclusions of quantitative analysis were added to the abstract and conclusions. (to see the abstract and the conclusions)

2. The language in this manuscript needs to be further improved and polished.

Answer: Thanks for your advice. We invited professionals to polish the paper.

3. The clarity of Figure 5,6,7,8,9,10,11 and 12 are needed to be improved.

Answer: Thanks for very much. Simulation experiments have been implemented again to ensure that the clarity of the figures meets the requirements. (to see the Figure 5,6,7,8,9,10,11 and 12)

Round 2

Reviewer 2 Report

  1. For the abstract , avoiding the express 'in this paper ', it should be more objective for paper published.
  2. From line 10 to line 12 , 'in the off-line phase,... In the real- time control phase...', it is abscent with some connections to  convince  readers there could be established without any priori knowledge information about   environment.
  3.  As mentioned in line 62, Secondly, it was difficult to find the logic , for that between this 2 reason , there are some many sentence, if possible , seperating these to paragraph.
  4. Between line 80 to line 83, it is still not so clearly expressed how to train during off-line phase , or it need no any train procedure , for training a general view point for Neural Networks. In which way or using what kinds of data for training.
  5. For the line 144 to line 146, 'When external disturbance occurs, 
    the controller can adaptively change the control law to ensure that the running state of the underwater vehicles is still on the reference trajectory' , it was curious how to adaptively change for such non-linear system, with what kind of information to deal it.
  6.  For the question ' In the line 328-330, the average optimization time RBF-NMPC is 0.0144 s and that of LMPC is 0.0076 s., how to measure and get this time, as the sample period said as 0.05s, in the line 319'.For my knowledge,since  you set the sample period as 0.05s, at least it was needed some  sample periods to adaptive to the environment . It could not be less than sample period as 0.05s. 

Author Response

1.For the abstract, avoiding the express 'in this paper ', it should be more objective for paper published.

Answer: Thank you for your advice. The express of abstract has been modified more objective. (to see the abstract)

2.From line 10 to line 12, 'in the off-line phase,... In the real- time control phase...', it is abscent with some connections to convince readers there could be established without any priori knowledge information about environment.

Answer: Thank you for your advice. The connection between the two phases has been added. (to see the line 13)

3.As mentioned in line 62, Secondly, it was difficult to find the logic, for that between this 2 reason, there are some many sentence, if possible, separating these to paragraph.

Answer: Thank you for your advice. The two main difficulties in using the nonlinear model predictive control has been separated into two paragraphs. (to see the fourth paragraph and fifth paragraph of the introduction)

4.Between line 80 to line 83, it is still not so clearly expressed how to train during off-line phase, or it need no any train procedure, for training a general view point for Neural Networks. In which way or using what kinds of data for training.

Answer: Thank you very much. This part has been rewritten. Training details for offline phase have been added. (to see the line 82-89)

5. For the line 144 to line 146, 'When external disturbance occurs, the controller can adaptively change the control law to ensure that the running state of the underwater vehicles is still on the reference trajectory', it was curious how to adaptively change for such non-linear system, with what kind of information to deal it.

Answer: Thank you very much. The underwater interference is decomposed into an interference force on each degree of freedom, which causes the state of the underwater vehicle to change. Therefore, it can be considered that the state information of underwater vehicle contains the information of external interference. The information can be used to adapt to the environment. (to see the line 150-156 and line 207-213)

6.For the question ' In the line 328-330, the average optimization time RBF-NMPC is 0.0144 s and that of LMPC is 0.0076 s., how to measure and get this time, as the sample period said as 0.05s, in the line 319'. For my knowledge, since you set the sample period as 0.05s, at least it was needed some sample periods to adaptive to the environment. It could not be less than sample period as 0.05s.

Answer: Thank you very much. In each sampling period, after the new sample data is collected, the network parameters are updated once by the error between the new sample data and the network prediction output. So the optimization time is not very long. (to see the line 216-218)